# “Knowing I Had Someone to Turn to Was a Great Feeling”: Mentoring Rural-Appalachian STEM Students

**DOI:** 10.3390/bs14010075

**Published:** 2024-01-20

**Authors:** Henrietta S. Gantt, Leia K. Cain, Melinda M. Gibbons, Cherish F. Thomas, Mary K. Wynn, Betsy C. Johnson, Erin E. Hardin

**Affiliations:** 1Department of Educational and Psychological Studies, University of Tennessee, Knoxville, TN 37996, USA; 2Department of Educational Leadership and Policy Studies, University of Tennessee, Knoxville, TN 37996, USA; 3Department of Psychology, University of Tennessee, Knoxville, TN 37996, USA

**Keywords:** mentoring, STEM, rural Appalachia

## Abstract

Post-secondary students benefit from mentorships, which provide both emotional and academic support tailored to the unique challenges they face. STEM students, and, in particular, those with historically marginalized identities, have unique strengths and face distinct barriers that can be ameliorated by careful, knowledgeable, and well-situated mentoring relationships. With that in mind, we conducted a narrative case study with 10 rural-Appalachian STEM majors enrolled in an NSF-funded mentoring program, intending to collect stories of their impactful experiences with their mentors. We utilized the narrative reconstruction process, and, in so doing, identified five major themes related to the importance of mentor assignment and the impact of mentors’ characteristics and skills related to empathy, consistency, active listening, and teaching. We situate our findings within the existing literature and provide implications for scholars and practitioners who work with mentoring programs dedicated to working with Appalachian communities.

## 1. Introduction

Students from the Appalachian region, a 13-state area with over 26 million residents, face unique career challenges that are specific to the region. The Appalachian region is characterized by egalitarianism, responsibility to family, independence, and interconnectedness [1,2]. Within Appalachia, the regional culture has a hallmark “we take care of our own” outlook ([1], p. 153). The historical influences of farming and coal mining as primary industries in the region have created lasting influences on family values, community, and interdependence [1,3]. The recent declines in these industries have contributed to challenges in the region, including barriers related to higher poverty rates and lower rates of college enrollment and completion [4]. Only about one-quarter of Appalachian adults have completed a college degree, which is lower than the 33% national rate [4]. Those from rural Appalachia are even less likely to earn a college degree (17%), while being more likely to live in poverty (18.9% [4]). Those who successfully transition to postsecondary education may need support to successfully navigate the unfamiliar world of academia.

Post-secondary STEM students encounter various challenges related to sense of belonging [5], academic stressors, mental health issues, and other hurdles throughout their collegiate programs [6,7]. Students with historically marginalized identities and those from underprivileged backgrounds often face unique and increased challenges as compared to their more privileged peers [1,8,9], while remaining underrepresented within much of the historical literature on higher education and psychology [7]. Appalachian students often face particularly unique challenges and triumphs based on their geocultural background [6,10,11]. Furthermore, both students who are of lower socioeconomic status (SES) and first-generation college students (FGCS) are traditionally underrepresented within STEM fields [12,13,14]. With this in mind, we sought to explore the experiences of STEM majors from rural Appalachia and the impacts of having a formal mentorship utilizing core helping skills.

Efforts to support rural-Appalachian students in academia would benefit from a culturally sensitive approach that considers the values associated with the cultural context. Hall et al. (2016) completed a meta-analysis on psychological interventions and found that those that were culturally adapted, or those that centered participant culture, were more effective than strict, unrevised, manualized approaches [15]. Their review included various culturally adapted interventions, including those using CBT, psychoeducation, family therapy, and school-based social development. Interventions were adapted for different cultural and social identities, including Latinx, African American, and Asian children, adolescents, and adults, and produced better outcomes than non-culturally adapted interventions. The most impactful interventions consider both empirically supported interventions and the unique strengths and challenges of the group being served [16]. Similarly, two recent scoping reviews indicated better outcomes for Latinx with substance abuse disorders, and improvement in depressive and anxious symptoms in diverse youth populations when interventions were culturally adapted [17,18]. The authors of both scoping reviews recommend adapting interventions for the specific and unique needs, values, and lived experiences of the target population. Together, these reviews offer initial evidence that cultural adaptation is an important component of effective psychosocial programming.

Appalachians differ from most other residents of the United States in that they tend to be collectivists—focusing on family and community needs over individual concerns [10,19], making prosocial characteristics like neighborliness, a sense of humor, humility, and modesty highly valued [20]. Appalachians tend to build very close-knit social networks while also valuing localism and a love for the land [10,20]. This sometimes makes the transition to postsecondary education particularly challenging for Appalachian students who may be leaving that support system for the first time. Counterbalancing the importance of family and community is a common distrust of outsiders or members of other cultural groups, one which results from historical exploitation [21]. Taken together, this indicates that building a social network may be particularly important for Appalachian students, though this process might be hindered by the hurdle of distrust. 

In addition to family and community, Appalachians tend to highly value qualities like self-reliance, independence, and hard work [20]. These values often contribute to a bootstrapper mentality, or a belief that they will be able to overcome challenges through their mettle and effort [10]. It may seem that this contrasts with their collectivism, though some scholars have theorized that strong family ties support the development of self-reliance [20], while others posit that Appalachians may avoid asking for help out of shame or to avoid being a burden [19]. Despite believing in their ability to overcome, many Appalachians exhibit fatalism or an external locus of control, meaning that they believe many outcomes are out of their control [21]. In combination, these two belief systems support both a present-oriented mindset, meaning that Appalachians tend to deal with problems as they arise [21], and a positive future mindset, in that they tend to believe that things will “work out” [10]. These complex mindsets are important to consider when working with Appalachian students. 

As many rural-Appalachian students come from low-income families and have limited exposure to academia before their immersion into the college experience, early and ongoing support in the forms of academic assistance, mentorship, and financial aid offers these students the most success in college [6,22,23]. The Committee on Effective Mentoring in STEMM defined mentorship as a professional relationship that promotes positive growth through providing both career and interpersonal support [24]. Mentorship specifically contributes to success in college by helping students navigate resource supports that they may not know to exist [25,26,27]. These students also benefit from mentors who provide holistic support, not only seeing the students as students needing academic support, but also understanding the students within their social and emotional contexts and, as well, providing relational support [26,27,28] which aligns well with Appalachian cultural values. 

For low-income, first-generation, and other underrepresented students, mentorship support is associated with higher overall GPA and the increased chance of continuing education with graduate-level enrollment [29,30]. Specifically, among rural-Appalachian students, many embody a cultural value of respect for authority and, characteristically, would not openly question faculty or others in positions of power [2]. As such, mentoring relationships help students to understand where they can ask for more support from professors and others within the institution. The mentor may teach the Appalachian student about resources on campus that offer support or encourage them to meet with their professors during office hours.

Ideally, mentorship offers holistic support that not only assists students academically, but also provides social and emotional encouragement. Mentors serve as role models and provide career-related support [7,26,27,28]. The literature related to mentoring most often considers the sense of belonging as a measure for evaluating mentor-program success, as mentor relationships encourage students’ sense of belonging on campus [7,27]. Good mentors provide psychosocial and academic support by showing warmth and concern for emotions, and by serving as positive role models [7,26,27,28]. Relationships in which mentees had similar backgrounds, identities, or experiences to those of the mentor also tended to be positive [7,29]. These commonalities helped students to feel a sense of connection. In evaluation results of mentorship programs, mentees indicated greater satisfaction with mentors who attended to the students’ cultural, social, and emotional needs [7,29]. These needs include understanding the student’s background and experiences, engaging with the student about their personal life outside of the academic experience, and “checking in” with the students about how they are feeling and adjusting to college. Mentorship that was culturally, socially, and emotionally relevant had a positive impact on students’ development of goals (personal and career), sense of belonging, and connection [7,29].

However, limited information exists related to the type of training needed for mentorship, and mentor levels and relationships with students vary by study, with the mentor sometimes also serving a dual role as a peer, faculty, or school staff member. The current literature on mentoring relationships with undergraduate students does not provide a consistent model of who should be providing mentorship or what training is needed. Additionally, little information is provided on how to tailor mentorship for students from specific underrepresented backgrounds or in specific undergraduate programs.

Nationwide, students from lower-income households and rural-Appalachian students are less likely to major in STEM fields; fewer students from rural communities voice interest in STEM majors, and, for those who do choose to pursue them, they struggle to persist to graduation [31]. Thus, on top of the inherent academic challenges associated with STEM degrees, rural-Appalachian students may not only have fewer college-educated role models, but also fewer STEM-educated role models, contributing to a lack of access to information and support outside of college. Their underrepresentation in STEM may also add challenges to their sense of belonging in STEM fields. Thus, mentorship is especially important for rural-Appalachian students interested in STEM.

As noted above, research has shown that mentorship is an effective intervention for student success, helping students in their development of confidence, sense of belonging [32], personal and academic growth [7,25], and research-skill development [29]. STEM students also benefit from mentorship, as it aids in research identity development, and this identity encourages students to continue their education post-bachelors-degree [30]. Rural-Appalachian undergraduate STEM students have specifically identified mentorship as being impactful for their social, personal, and academic success [6]. Therefore, more research is needed to provide knowledge on how to effectively mentor this population.

Research on mentoring repeatedly calls for more studies on the mentor characteristics that lead to effective mentoring (e.g., [7,33,34]). One recent study [35] attempted to answer these questions using retrospective reflections from underrepresented students who had participated in a program designed to foster retention and success in STEM. The authors found that students who reported having a faculty mentor had an increased sense of belonging and self-efficacy and increasingly positive attitudes toward STEM and science than students without a mentor, but also that greater mentorship support (defined as psychosocial support, career support, and psychological closeness) was key to these psychological outcomes. However, this study leaves other unanswered questions, like the specific mentor behaviors that communicated this support. Crisp et al. (2017) highlighted the need to further explore what aspects of mentoring are most impactful, particularly with respect to the creation of effective mentoring relationships [36]. To address this gap in the research, we interviewed low-income, rural-Appalachian STEM majors about their mentoring experiences.

Although scholars have repeatedly established that relationships with high-quality mentors can positively affect the development of these students (e.g., [6,10,12]), and many universities have implemented mentoring programs to support underrepresented STEM students, mentoring activities vary widely and primarily focus on academic, social, or career-specific support [7,12,23]. Additionally, few researchers have examined how mentees perceive their relationships with their mentors [37]. Thus, we need further scholarship concerning the experiences of underrepresented students that are most impactful within mentoring relationships to better understand how to support this group. To that end, we utilized narrative case-study methods to explore the experiences of 10 STEM majors from low income, rural-Appalachian backgrounds in engaging with their assigned mentors in a four-year undergraduate support program at a public, four-year, research-intensive university situated within the Appalachian region of the U.S. We specifically pursued narrative interview practices to honor the long-standing, deeply cherished, culturally rooted tradition of storytelling, which is prevalent across Appalachia [6,38]. We asked participants to tell us stories of experiences with their mentors in order to answer the following research questions: (1) What qualities in a mentor are most impactful for STEM majors from low income, rural-Appalachian backgrounds? (2) What interactions with a mentor are most impactful for STEM majors from low income, rural-Appalachian backgrounds?

## 2. Materials and Methods

We conducted a narrative case study to capture the lived experiences of our participants through storytelling [39] and to honor the culturally grounded practice of storytelling within Appalachian cultural spaces [6,38]. We conducted narratively oriented, semi-structured interviews with each of our ten participants in order to provide a light structure for data collection while holding space for each individual to share their own unique stories, without providing guidance as to a certain theme. For example, we prompted, “please tell me a story about a time that you met with your mentor”, tasking participants with the freedom to choose their own direction, story structures, and portrayals of themselves and others, without leading each participant to a predetermined outcome.

### 2.1. Case Description: The ASPIRE Appalachian Mentoring Program

The ASPIRE program (NSF 1643393) provides support programming and funding for high-achieving, low-income, students from rural-Appalachian backgrounds majoring in an arts and sciences STEM field (e.g., chemistry, biology, physics, and neuroscience). Overseen by faculty from psychology and counselor education, the program provides scholarships, program-specific courses, research opportunities, and ongoing mentor support for these ASPIRE Scholars. Graduate students from psychology and counselor education act as mentors for ASPIRE Scholars and are vital components of the ASPIRE program. In addition to providing one-on-one mentoring, these graduate student mentors, known as AMPs, also teach the program’s courses, which cover topics such as researcher identity development and planning for graduate school, within a scholar-cohort context.

Over the course of the program, ASPIRE has employed eight AMPs. Of these, three were male and five were female; all but one identified as white, and many returned for repeated years of service. AMPs are matched with four to eight ASPIRE Scholars as their mentees and meet with them a minimum of two times per semester outside of their ASPIRE courses. Additionally, most mentees also maintain regular, ongoing contact with their AMP via email and texts. AMPs are trained in helping skills such as active listening, empathy, and strengths-based communication, in addition to their training in academic advising. AMPs also engage in training focused on the best practices for working with rural-Appalachian and low-income students, and all participate in research projects associated with this population. Furthermore, the two faculty coordinators serve as supervisors and provide weekly supervision of AMPs to discuss student progress, answer questions, and address any concerns the AMP or supervisor may have. In training our AMPs in these areas, we hope that they approach mentoring as a combination of supportive listening and expertise in how to address academic, social, and personal challenges.

### 2.2. Participants

Upon receiving IRB approval, we sent recruitment information about the study to each of the 30 ASPIRE scholars participating in the program, 10 of whom volunteered to participate in the study and completed a one-hour interview. Of our 10 undergraduate STEM students, six identified as women and four as men; nine identified as non-Hispanic white, while one participant identified as Asian. Two participants, Kat and Sam, were also completing a second major in Anthropology. Unsurprisingly, seven participants identified as a First-Generation Student (FGS)—a statistic supported by other scholars who have worked with Appalachian collegians [1,10]. We have presented individual demographics for each participant in Table 1; each participant is listed using a pseudonym that they chose for themselves.

### 2.3. Data Analysis and Quality

Upon completion of the verbatim transcriptions of each interview, we engaged in the narrative reconstruction process [39,40]. First, we identified story elements throughout each transcript; for example, when interacting with transcript data wherein a participant was directly asked to “share a story about a time where your mentor was particularly impactful”, we “restoried” their responses into chronological order with details gleaned from across the remainder of the interview (e.g., later references back to an earlier story), as well as any “side stories” that cropped up about other instances along the way [39,40].

Upon completion of the narrative reconstruction process, we divided the 29 vignettes amongst our team members, with each vignette being assigned two readers. Each reader then read through their assigned vignettes and generated a list of themes that they felt best represented the stories. Thorne et al. (2004) described this form of thematic analysis process as *inductive meaning making*, an interpretive process that identifies themes across participants while also maintaining the unique qualities inherent to the individual experience ([41], pp. 3–4).

We then met to compare our findings and negotiate potential themes [40,42]. Fortunately, our themes were all very similar, with slight variations in labeling (e.g., “consistent” and “always shows up” were combined into “consistent”). This process allowed us to engage in an inductive, co-constructed meaning-making process [42] that we felt best represented our data. We explore our themes and provide demonstrative vignettes in our findings section.

Scholars have long struggled around the diverse and varied constructs that we embrace as (and rarely agree upon the validity of) multiple types of quality indicators [43,44,45]. Therefore, in designing our study and choosing the quality criteria by which we would judge our decisions, we met as a team, negotiated our philosophical values and paradigmatic commitments, and chose to primarily pursue evidence of credibility, rigor, critical reflexivity, polyvocality, significance of contribution, and the centering of our ethical decision-making. We define and detail each of these concepts in Table 2.

Our research team consisted of people with diverse identities, perspectives, experiences, and philosophical commitments. In order to maintain open and effective communication, we engaged in reflexive journaling and group conversations, and frequently “checked in” with each other throughout the data collection and analysis process in order to reflect on our subjective beliefs and values in order to promote rigor and transparency in our coding process [47]. Through our varied lenses, we were able to engage in meaningful negotiations around meaning-making throughout the inquiry, ultimately resulting in a more holistic and nuanced understanding of our findings [6,43,47]. Philosophically, two of our team members identify as critical interpretivists, two as constructivists, and two as critical realists (see [51]; the seventh author did not engage in interviews or data analysis). Three of our team members were actively involved in the program as mentors and faculty advisors, while the other team members were external to the program, allowing both *insider* and *outsider* lenses to engage in our analyses. Additionally, all team members have had prior experience working within Appalachia and currently live within the region, we each identify as white women, and primarily come from the Southeastern (*n* = 4) and Midwestern (*n* = 3) U.S.

## 3. Findings

We organized our findings into one overarching theme which we use to frame our discussion, namely, *Mentor Assignment*, and identified four themes representative of mentoring qualities that participants shared as being impactful to their experiences; specifically, a good mentor is (1) *empathetic*, (2) *consistent*, (3) *a listener*, and (4) *a teacher* (see Figure 1). In this section, we define and describe each theme and its subthemes and provide vignettes and quotes from our participants to demonstrate.

### 3.1. Overarching Theme: Mentor Assignment

As we interviewed each participant, it emerged that a notable feature of each individual’s stories revolved around experiences either (a) bonding with their mentor over many years or (b) having their mentor changed as a result of mentors graduating or otherwise transitioning out of the program. For example, Emma shared the following vignette about how she eventually grew to depend on her mentor Jane:

“At first–freshman and sophomore year– I resisted going to my mentor, Jane, because I didn’t want to seek help. We were supposed to meet twice a semester, but I just didn’t know her yet. I was having some serious roommate problems that I needed help with….she would, like, *torture* me, and my mental health was starting to decline. It was hard for me because it was my first year away from my family–away from everything. I was really resisting Jane, but I was required to meet with her, so finally at the end of the semester, I kind of just *unloaded* on her and she told me how I could deal with her. Not necessarily to *change* my roommate, but to help me *cope*. Jane suggested, ‘well maybe you should exercise or write it in a journal–or maybe if you can’t do your homework in your room, you should go to the library.’ She also recommended peer mediation and listening to music with my headphones. We just found ways to calm down. She’s always helped me find ways to relieve my stress. I finally started to trust Jane.

More recently, I was considering grad school and it was making me really anxious, I mean–I’m a first-generation student–I didn’t think I was going to get in. Her support was imperative because my family did not like the idea. They were like, ‘you’re not gone do well,’ ‘you’re not gone get in,’ ‘how you gone get a career after your bachelor’s?’ So, without Jane, I don’t think I would have been able to go through the process–it was just *so stressful* with COVID and stuff. Jane stayed calm and talked me through the entire process. We worked on my personal statement a lot and she helped me come up with a plan. She gave me so much support and motivation to continue through the process. Because of her, I was able to get accepted into a master’s program.

It’s been really encouraging to have her, because it feels like having an older sister to help guide me. I meet with her, like, four times a semester now. Although I resisted her at first, now I see her more often than my other friends and supportive folks–I’m actually meeting with her after this meeting to talk about getting accepted to master’s programs! She’s probably the best thing the Program has given me. She became my main resource when I’m stressed out. I always feel like I have things stacked against me, and I always just push through even if I don’t know what it is that’s pushing against me. Knowing I have someone to turn to and to is just a great feeling. I was very unsure at first, but once I realized that I needed the support, and Jane was there to give it, it did change, like, my entire college experience.”

Although Emma benefited greatly from Jane’s consistently affirming presence, Darien’s experiences were quite different, due to his mentor, Eric, moving on from his role in the program. Darien struggled with “two or three different mentors”, positing that “it’s harder to start [the relationship] from scratch–I’m not as personally connected to the [other mentors].” When describing this difficulty, Darien shared, “Eric knew a lot about me, I guess… so he was able to build off of that stuff versus with new peer mentors, it was kind of hard [to begin a new relationship].” Similarly, Sam changed mentors during her program of study, and “didn’t have the same connection with” her new mentor, compared to that with her initial mentor, though she noted that the newer mentor was “still really, really helpful–just in a less personal, different sense.”

Each participant’s stories were framed by the relationship that they shared with their mentor(s); although some, like Emma, grew incredibly close to the mentor who spent years by their side, others, like Darien and Sam, noted struggling due to the lack of consistent mentorship from a single source.

### 3.2. A Good Mentor Is Empathetic

One mentorship quality that was commonly spoken of as being important to the effectiveness and impact of the mentoring relationship was empathy. An empathetic mentor was someone who (1) showed that they cared about the mentee, (2) demonstrated awareness and understanding of what their mentee was experiencing, and (3) was able to give honest, relevant, and helpful advice. In the following vignette, Daisy shares how Jane’s awareness, understanding, and validation of what Daisy was experiencing made Daisy feel cared for.

“Some of the main times that I felt appreciated or valued was during my mentor meetings. I had to meet with Jane twice a semester and kind of update to tell her what was going on. And as routine as it sounds, it was nice to do regular check in with the same person all throughout four years. She was always like, ‘Hey, how was that one thing? How’s this going? How’s blah, blah, blah.’ And so I think that that was nice, because it was appreciated. I felt appreciated that they remembered stuff about me.

It kind of gives me a sense of reassurance, I guess, in terms of like what I’m doing. Because sometimes in college, I do feel isolated. And you do feel like no one understands what I’m going through. And so it is kind of nice to just talk to someone and have them be like, ‘okay, I hear you, that is a hard thing. And I’m proud of you. And you’re doing this’ and you’re like, ‘Okay, okay, so nice.’”

According to Cassiopeia, their mentor was able to use the understanding and awareness of their mentee’s experience to anticipate the mentee’s needs and offer support accordingly:

“And so, I guess it was nice how she was like, aware that graduate school was kind of a foreign concept to me, because I hadn’t really reached out to anyone else about it. I felt a little awkward about it because I felt like I should know more, but luckily she got that. So, she was able to give me a really nice, basic introduction of the things that I should know and told me what things I should be thinking about and maybe what questions I should be asking some of the people in the physics department to help me out.”

Cassiopeia’s mentor was aware that Cassiopeia did not know very much about graduate school and understood how this impacted Cassiopeia’s feelings and actions. Based on this, the mentor was able to tailor the support to give Cassiopeia the knowledge and encouragement Cassiopeia needed to build confidence. Cactus’ mentor was able to offer honest and critical advice based on the breadth of the mentor’s awareness.

“Having my mentor follow up across meetings and ask specifically about things helped me reflect. She didn’t tell me to do whatever makes me happy, or give enabling advice. She gave me good, critical advice to help me analyze my thoughts and feelings and really think it through. Some of my favorite moments have just been us going through all my life crises of what I wanted to do after college and having her offer advice and ask really good questions to help me think through them and figure things out.”

Having a deep understanding of Cactus’ experience enabled her mentor to give quality advice in a way that both was helpful and communicated efficiently to the mentee.

### 3.3. A Good Mentor Is Consistent

In our interviews, the participants expressed a common theme, namely, that of a good mentor being consistent. We defined “consistent” as being actively involved in their mentee’s lives and being a constant source of support. A consistent mentor (1) was available to their mentees, (2) valued their mentees, (3) was involved with their mentees, and (4) was a constant in their mentees’ lives or college experiences. Below, Cactus shared her experience of benefitting from the consistency of her mentor, Jane, and how Jane was able to remember all of the little details. Because Jane was a consistent presence in Cactus’s life, Cactus was able to have more opportunities to reflect and to have a trusted person to consult with during her college journey.

“We do the AMP meetings twice a semester and I have had Jane the entire time. I feel like having Jane specifically watch me going from where I was to where I am now has been really nice. She references her notes from past conversations to check up on little things. In one meeting I was telling her I hate it here because of x, y, and z and in the following meetings she followed up on all of those things. Having the continuity between it all has been really good. By reflecting on my last semesters and my whole experience, she was able to give me really good advice.”

Similarly, other participants shared how the consistency of their mentors offered more opportunities to access resources and to have a trusted person whom they knew they could reach out to. Cassiopeia described the benefits of the availability and involvement from their mentor, saying “anytime I’ve needed something, or if I have miscellaneous questions about research, or any sort of stuff, I’m always able to text her and she’s available and helpful. Even recently, she helped me take advantage of the summer research funding [offered by the program].” Similarly, Kat described how the constant involvement of their mentor served as a tethering force, saying, “I can go to my mentor for anything. I’ve even asked him to be my reference for a job. You know, campus is huge–but I have someone I’m always tied to. I set goals and he follows up.” Participants benefited differently in their experiences as to the theme of consistency, but repeatedly highlighted how consistency from their mentors facilitated more impactful relationships.

### 3.4. A Good Mentor Is a Listener

Each of our participants shared stories of their mentors being easy to talk to, and ultimately making each mentee feel truly heard and as though the mentee could safely share their deepest thoughts and insecurities. The theme of being a listener is distinct from the quality of being empathetic, as it describes the intentional act of creating space to wholly listen without guidance. Participants described benefitting from this intentional act of listening because they were able to relieve stress through venting and felt that their stories were valued because their stories were heard and understood by their mentor before any guidance was provided. For example, Darien shared the following while reflecting about his relationship with his mentor, Eric.

“I struggled a lot with the transition to campus during my freshman year, so having someone to talk to was really helpful–I could talk to Eric about everything. He was phenomenal. I was so sad to see him leave. I’ve never really had someone to talk to like that–he really felt like an actual, objective person that I could open up to. I talked to him about a lot of stuff. I would always come to our meetings thinking, ‘yeah, I’m just gonna get through whatever questions he asks,’ and then I would end up staying over our planned time just talking about other stuff, my personal life, and whatever else was going on. I always felt really good after talking to him–just from having someone to talk to. I was able to talk to him just about, just–about anything.”

Similarly, Emma shared that her mentor Jane “became like [her] diary”, eventually providing advice to help her “avoid conflicts and focus on school stuff.” She described how the experience of “unloading” on Jane also helped her to build trust so their relationship could grow over time. Each participant shared that their mentor’s ability to listen deeply aided their ability to make both intra- and interpersonal connections throughout their lives, ultimately impacting their academic and personal success.

### 3.5. A Good Mentor Is a Teacher

While our participants described the benefits of just being heard through the intentional act of listening, they also described the unique value of their mentors “teaching” them about skills and resources to help them succeed. Participants described how their mentors taught them new skills and shared resources so they could seek additional help from other sources when appropriate. When participants learned specific study or time-management skills, they were able to achieve more academically and navigate college with more ease. Luigi described how his mentor, Chris, shared multiple skills and techniques to help him through his freshman year,

“Chris, my Mentor, showed me how I could do a little better in school. The biggest thing he helped me with was time management. I didn’t have nearly as much homework in high school as I did here and I sort of just got drowned in all the homework and due dates. He encouraged me to use a planner and taught me how to pick apart a textbook and that was really helpful and I still use that stuff. He always made sure to let me know that it was okay to mess up sometimes and that I was still learning. Then, last year, my mentor changed to Olivia and we had a meeting where we discussed getting on top of things and reaching out to people. Specifically labs and ways to communicate with them and how to sell myself. That was probably the second most impactful one to me.”

Luigi was able to learn helpful skills from both of his mentors. Chris taught him skills in time management and studying, while Olivia taught him skills related to communication and networking. These skills helped Chris navigate his experience in college and be more prepared in the future.

Beaker shared a very similar experience. He was also having difficulty adjusting to the workload, as compared to high school. But his mentor, Shane, helped with time management: “He introduced me to some things that he does. He showed me the way he would schedule his day, or at least a technique that he knew about… and I still use it today. I have a planner that I write exactly what I need to do for the whole week.” Beaker also shared a story of another time he started to struggle academically, and Shane helped by connecting him to the right resources.

“I feel like he’s always provided me with things I needed or things I didn’t know I needed. He always provides good insight. One time, I was really struggling academically and our meetings helped a lot. First, he helped me destress some, then he gave me some contacts to reach out to and provided me with resources and things that would get me through it. Like SI sessions (tutoring sessions related to specific classes) and stuff that weren’t really announced in the class, but apparently existed. So it was good to have that outside source that he provided. I mean, sure, there’s resources out there for you, but you have to be looking. It’s nice for someone to just hand them to you.”

Mentors demonstrated support by both teaching tangible skills and connecting participants to resources. Tangible skills, mostly time-management and study skills, helped participants move through challenges, find greater success, and balance their many obligations. Connection to other resources allowed mentees to obtain more forms of catered support based on their specific needs.

## 4. Discussion

In this study, we explored how low-income, rural-Appalachian STEM undergraduates experienced mentoring as part of a comprehensive student support program. Though the ASPIRE program offers multi-faceted support to participating students, each of our study participants argued that the AMP mentoring program was the most impactful aspect of the program. Participants described how the mentorship supported their behaviors in their majors, research, and social contexts.

In recent years, researchers have begun calling for culturally responsive mentoring, or mentoring that centers on the lived experiences of those being mentored [52,53]. Mentoring is a well-established practice used to support college students, but more information about how to best provide mentorship is sorely needed [7,23]. In particular, a recent taskforce on how to effectively mentor STEM students highlighted the importance of centering cultural and social identities as part of the mentoring process [24]. Our findings help identify some mentoring aspects that are culturally relevant for rural-Appalachian college students. Mentors built relationships based on humility, empathy, and connectedness, considered the entire context of their mentee, and were consistent and accessible. Mentors that built relationships, considered the personal context of mentees, and had scheduled and regular meetings helped to build trust and improve the quality of their mentorship.

Our participants described how it took multiple semesters to build truly impactful relationships with their mentors, highlighting how adjustments to mentoring assignments led to less-personal, but still helpful, relationships. Other studies on undergraduate mentoring support this finding, highlighting that duration, or more time with mentors, increased mentoring relationship quality [26,30,37]. In these prior studies, the mentoring relationships were intentionally made stable over time. In contrast, although our mentoring program only required two meetings per semester, mentors typically had regular contact with students via program classes and informal chats over email or text. Participants appreciated having accessible and reliable mentors, citing ongoing contact (whether scheduled or student-driven) as positively impacting their mentoring relationships and overall academic outcomes.

Additionally, participants highlighted four qualities that were of paramount importance to their mentoring relationships, stating that a good mentor is empathetic, consistent, a listener, and a teacher. According to the National Academies of Sciences, most mentorship in STEM focuses on career guidance, in which the mentor helps the mentee develop their career path; while our participants appreciated this aspect of mentorship, the interpersonal and psychosocial support was far more valuable, providing evidence for the importance of culturally adapted mentoring. For these rural-Appalachian college students, effective mentors were aware of their mentees’ personal context. They listened to mentees’ stories and gave their time without judgment or rushing to solve problems. They remembered life-details and offered ideas without demonstrating a sense of superiority. In other words, culturally responsive mentoring for rural-Appalachian STEM majors centered on values such as egalitarianism, interconnectedness, humility, and burden-avoidance [1,2] to create strong relationships that were seen as helpful and supportive.

Some of these qualities were noted in prior research; for example, Crisp et al. (2017) reviewed prior literature on mentoring and proposed somewhat similar characteristics, including willingness to form emotional connections, providing encouragement, relating to students on their level, and demonstrating interest in the mentoring relationship [36]. Additionally, Leudke et al. (2021) added the importance of caring about both the personal and professional development of mentees as a necessary mentoring component [12]. Mentees described in the prior research often alluded to the characteristics noted by our participants. For example, students in research-mentoring relationships with faculty felt more supported when communication styles matched, and mentors were collaborative and responsive [29]. In a review of near-peer mentoring, mentees highlighted the helpfulness of supportive relationships that addressed both academic and personal needs [31]. However, the current study is one of the first to note these specific qualities from the perspective of mentees in general, and from rural-Appalachian, low-income STEM students in particular.

Our participants shared stories about specific occasions when their mentors helped them personally or professionally, but also described the general characteristics that helped them feel connected to their mentors. Although not specifically stated by the participants, it is clear that mentor training as to how to build and maintain strong mentoring relationships positively impacted them. Our mentors are trained differently from those in many other mentoring programs. AMPs received training on counseling skills from their graduate courses in counseling and psychology. Therefore, they will enter into mentoring with a strong set of helping skills. Then, they participate in academic-coaching training from the university’s academic success center. Finally, they engage in ongoing professional development and supervision in working with rural-Appalachian students, as provided by the program’s faculty coordinators. This combination of training and oversight addresses the need for prior mentor training and the culturally informed mentoring consistently called for by researchers [7,12,32]. It also highlights the helpfulness of prior helping-skills training, in which mentors are engaged in training for active listening, relationship and rapport building, and the core conditions of empathy, positive regard, and a nonjudgmental stance.

### 4.1. Limitations and Future Research

The limitations in this study are related to recruitment strategy, methodology, and COVID-21. Participants were recruited from one unique program, hindering the ability to gather more diverse mentorship experiences with this population. Future studies could address this by focusing on rural-Appalachian students from other universities. Furthermore, participants volunteered to be in the study, so not all program members’ stories were captured. Students may have volunteered because of their positive experiences with the program, which may have excluded some negative experiences held by students in the program. Limitations related to methodology may include the impacts of using a narrative inquiry and storytelling [37], as this may have limited some participants’ ability to share. Although we chose this approach due to the Appalachian tradition of storytelling, some students described the storytelling component of narrative inquiry as being challenging. One stated, “I’m kind of awful at telling stories”, while another shared that they would have preferred being asked more specific questions to answer. Therefore, the storytelling component may have hindered some of participants’ relations of shared experiences. In addition, the interviews were conducted during the COVID-19 pandemic, as were some of the experiences described by participants in the study. Participants’ connections to the interviewer may have been limited due to the online interviews. Also, their mentorship experiences may have been implicitly impacted by COVID in ways that may not have been explicit in participants’ stories. Future studies could consider the past impact of the pandemic to address these issues.

Although our goal was specifically to understand the unique experiences of predominantly first-generation rural-Appalachian STEM majors, the fact that our participants all came from one program at one university may limit generalizability to other groups. However, many of our findings were consistent with themes from other research on effective mentoring for other groups of students. Future behavioral-science research should explore the extent to which the unique combination of training and skills our mentors possessed (active listening and mental health first aid, academic coaching, cultural competence) would be as effective with other groups of students with different intersecting identities. Future research should also investigate more specific outcomes of effective mentoring, such as researcher identity, sense of belonging, and career goals.

Effective mentoring is already recognized as an important tool for promoting the success and successful behaviors of college students. Our findings confirm the importance of mentoring for an under-studied group, rural-Appalachian and predominantly first-generation STEM students. Our findings also highlight the importance of having mentors who can integrate academic success strategies with empathic listening and attention to mental health in a culturally sensitive context. Despite the fact that our mentors were not pursuing advanced STEM degrees, this combination of skills proved highly effective for these undergraduate STEM majors. If our results can be generalized to other groups of STEM students, this may suggest new approaches to effective mentoring for STEM students more broadly, which may in turn foster greater student success and well-being and reduce critical educational disparities.

### 4.2. Practical Implications and Conclusions

Our findings support practical implications related to behavioral sciences and creating supportive programs for secondary-education students, including mentor recruitment, inclusion, and training. Our findings are unique to the understanding of both mentorship and the behaviors useful for supporting students with intersecting identities, including being from rural Appalachia, holding a major in STEM, and being predominantly first-generation college students. These participants represent a unique intersection of development, culture, and career, an intersection with concurrent strengths and needs. Therefore, these suggestions for program development and mentorship training may better support the behaviors of students with similar identities and needs.

Previous researchers have suggested that students are more likely to be successful if mentors offer academic support and attend to their unique personal and social needs [3,27,28,29]. Our participants described ways in which their mentors demonstrated these qualities and techniques, connecting the skills directly to the students’ own attainment of success. Mentors can demonstrate emotional support by being empathic, consistent, and good listeners. For example, to demonstrate empathy, mentors can explicitly acknowledge what they appreciate and value in their mentee, respond to mentees without judgment, and demonstrate their awareness of mentees’ unique needs by offering unique advice and solutions to the mentees’ various concerns. Mentors can demonstrate their consistency by taking notes and referring back to them throughout meetings to show their awareness and support over time. Mentors should also be clear as to their availability while maintaining boundaries. Mentors should have predetermined accessible times to meet with mentees to ensure that the mentees are able to obtain long-term support. Furthermore, mentors can demonstrate that they are strong listeners by not rushing to find solutions for mentees, but, rather, by allowing time to let their mentee “unload” before they offer guidance. To demonstrate their strong listening skills, they can offer non-judgmental reflection and listen to the mentees’ unique concerns and unique emotions and thoughts surrounding their concerns.

The ASPIRE program is unique due to our focus on explicitly training mentors in academic coaching, allowing them to serve as teachers by sharing specific academic strategies and appropriate campus resources for students. Our results support the importance of having mentors who can provide information about on-campus counseling services, tutoring opportunities, and career or graduate-school guidance. Mentors can also “teach” students about study skills and coping skills to help them with academic and personal challenges. Participants’ mentors showed them how to navigate studying textbooks and time management as well as coping strategies and behaviors including journaling, mindfulness, and listening to music.

Our unique mentor training and findings on empathy, consistency, listening, and teaching imply that training research mentors, social mentors, or behavioral-science faculty for vulnerable populations and STEM majors should include training to pragmatically and emotionally support these students. The training should include active listening and understanding of the mentee’s cultural identities, campus resources, and study/time-management skills, as well as basic “mental-health first aid.” Helping professionals are well suited for these roles; therefore, programs recruiting mentors may consider utilizing graduate students in counseling, psychology, or social-work fields or hiring mentors with these backgrounds. Students may benefit the most from the combination of intentional listening and empathy, matched with appropriate guidance on campus resources, or unique skills to assist them with their unique challenges.

Knowledge and resources on SEL (social emotional learning) may be beneficial in training mentors to match the findings of this study, particularly those related to empathy and listening. SEL skills help both children and adults to recognize and manage their emotions, communicate effectively, and solve problems [54]. Recent research has particularly shown how effective SEL skills can be in helping undergraduate students with their academic success [54]. Therefore, professional development resources using SEL skills similar to those found through the *Transforming Education SEL for Educators ToolKit* [55] can be used to inform the training of mentors, attending to their ability to offer active listening skills, empathy, and problem-solving techniques.

Our participants each had assigned mentors who supported them throughout their college experience. Participants reported difficulty adjusting to changes in their mentor assignment. Statements from participants, such as, “it’s hard to catch this person up” emphasize the significance of the constant developmental and transitional changes encountered by college students as they enter and navigate college. Furthermore, researchers suggest that rural-Appalachian and first-generation students may have a tendency to distrust outsiders or those persons in positions of authority [5]. We found similar sentiments from the present participants in their stories. Other students reported that it was hard to recreate the connection with changing mentors. These findings suggest that inclusion of mentorship in supportive programming for vulnerable populations may be most beneficial to students if programs have long-standing contracts with mentors, faculty, or advisors so that they may better support their students’ developmental and cultural needs.

## Figures and Tables

**Figure 1 behavsci-14-00075-f001:**
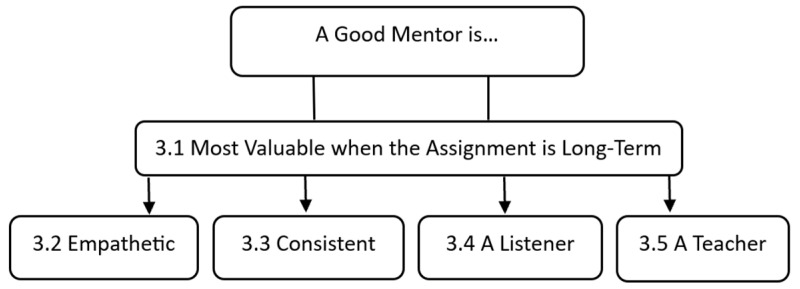
Findings.

**Table 1 behavsci-14-00075-t001:** Participant Demographics.

Pseudonym	Year	Major(s)	Race	Sex	FGS
Beaker	2	Chemistry	Asian	M	Y
Cactus	4	Neuroscience	White	F	Y
Cassiopeia	3	Physics	White	M	Y
Daisy	4	Physics	White	F	Y
Darien	4	Biochemistry, Cellular and Molecular Biology	White	M	Y
Emma	4	Neuroscience	White	F	Y
Kat	2	Neuroscience, Anthropology	White	F	N
Kirby	3	Neuroscience	White	F	N
Luigi	3	Chemistry	White	M	Y
Sam	2	Biological Sciences, Anthropology	White	F	Y

Note: We have chosen to define FGS as a student for whom neither a parent nor a legal guardian has completed a four-year degree or higher, as this is the definition used by the institution where the study took place.

**Table 2 behavsci-14-00075-t002:** Data Quality Considerations.

Data Quality	Defined	Evidenced
Centering Ethical Decision-Making	Scholars must prioritize ethical reasoning and the wellness of participants [46,47].	We obtained all relevant IRB permissions, obtained informed consent from each participant, and ensured that no participant was interviewed by a former mentor.
Credibility	Scholars must provide sufficient evidence to demonstrate that their interpretations of the data are appropriate [47,48].	We have provided vignettes and quotes from each of our participants and worked to ensure that they were represented equally throughout our findings and demonstrate the saliency of our themes.
Critical Subjectivity	Scholars must work to deconstruct their beliefs, biases, and subjectivities to understand how these lenses affect their approaches to the data [49].	As a group, we critically engaged our individual and group subjectivities and frequently checked in on each other’s biases to negotiate decision-making during analysis. We have provided more details about this process below.
Polyvocality	Scholars must consider whose voices are over- and underrepresented and seek to prioritize the voices of those who are silenced [49].	Our study population consists of underrepresented, erased, and often-silenced individuals from Appalachia.
Rigor	Scholars must provide a detailed, clear description of their design and methodology [47,50].	We have sought to provide transparent, clear details of our decisions and processes throughout this manuscript.
Significance of Contribution	Scholars must work to address theoretical, methodological, practical, and cultural gaps within the literature [47].	By completing this study, we seek to contribute to narrowing the gap present throughout the literature surrounding how to best support Appalachian students. This is further explored in our implications.

## Data Availability

The data presented in this study are available on request from the corresponding author.

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
