# Peer review of "“Knowing I Had Someone to Turn to Was a Great Feeling”: Mentoring Rural-Appalachian STEM Students"

_behavsci, 2024, doi:10.3390/bs14010075_

Round 1

Reviewer 1 Report

Comments and Suggestions for Authors

The paper investigates the mentoring experiences of STEM students in the rural Appalachian region, a topic that is not widely explored in current literature. This focus provides a unique perspective in the field of education and mentoring. The study also significantly contributes to the scholarship by shedding light on an understudied group in STEM education research (i.e. rural Appalachian, low-income, and first generation students). It provides valuable insights into the specific needs and challenges faced by rural Appalachian STEM students in mentoring relationships. The study's findings have the potential to influence educational practices and policies, particularly in addressing educational disparities in rural areas. This contribution is particularly important as it addresses a gap in the literature and offers practical implications for educators and policymakers.

In terms of quality, the paper is well-structured, with a clear introduction, methodology, results, and discussion. However, there is potential for improvement in the organization of the findings and discussion sections. Presenting the results in a more comprehensive but focused manner, would enhance the paper's readability and impact. Additionally, linking the discussion more directly to the results would help in making a stronger, more cohesive argument.

The paper engages well with a range of relevant sources, demonstrating a thorough understanding of the existing literature on mentoring in STEM education. To further strengthen this aspect, the authors could incorporate more recent studies, especially those published in the last few years. Adding international perspectives or studies from different educational contexts would also enrich the paper’s engagement with the broader scholarly conversation. To strengthen the paper's argument, the paper could include a more critical examination of its findings in relation to existing literature. While the paragraph in lines 496-508 attempted to accomplish that, the related literature were only framed in terms of the likeliness of their findings to the current study. A more critical comparison would benefit the discussion section. Moreover, discussing potential limitations in greater detail and suggesting areas for future research would also enhance the paper’s academic rigor.

Overall,
this paper is of high merit. It tackles an important and under-researched area in STEM education, focusing on a demographic that is often overlooked in academic studies. The insights provided are not only academically valuable but also have practical implications for improving mentoring practices in rural and underserved regions. The study stands out for its relevance, depth of analysis, and potential to inform future research and practice in the field.

Author Response

Please see attachment. Thank you for your feedback!

Reviewer 2 Report

Comments and Suggestions for Authors

This is an interesting qualitative study examining the narrative experiences of mentoring of STEM majoring students from Appalachia. The authors have sufficiently engaged with recent literature, have attended to the issues of trustworthiness, and have provided a good description of the methods. Overall, I have a good impression of this paper but would like to see some more attention paid to the following issues:

1.      Some reordering is needed in the introduction section. Specifically, the paragraph in line 36 should come at the bottom of the introduction in line 181 after the authors have clearly outlined the context of the research.

2.      Although it is implicitly mentioned in the text, the introduction would benefit from a clear definition of mentorship.

3.      The introduction section should conclude with a summary of the information, an outline of the practical and theoretical “gaps” in knowledge and concrete research questions. As it is now written, the introduction leaves the reader without clear focus on what this research is about.

4.      The ASPIRE programme description should be in the introduction rather than the methods section.

5.      The authors use quotations form references in some places, but this is not a thesis. Hence, I think that these quotations should be summarised rather than presented in carbon-copy from the initial sources (e.g., lines 244-247).

6.      Are these participants college students?

7.      There is a typo in line 304.

8.      I would like the authors to consider a bit more about the distinction between the themes “A good mentor is a listener” and “a good mentor is empathetic”. Being empathetic includes “empathetic listening”, that is, being attentive, trustworthy, non-judgmental.

9.      Finally, the discussion section feels like a repetition of the key findings. It is very descriptive but it is missing the key structure where the findings are consistently compared and contrasted with previous studies. Also, I suggest the authors try to highlight more the innovativeness and the originality of their approach.

Author Response

(The authors gave the same response as above.)

Reviewer 3 Report

Comments and Suggestions for Authors

Thank you for the opportunity to review this work on an important topic. The manuscript focuses on supporting STEM career development among underserved populations. Overall, the authors described relevant literature, methods, findings, and discussion in a meaningful and organized way. Detailed comments are provided below.

The authors discuss the importance of using a culturally sensitive approach in Lines 70-82. The authors describe it as an approach that “takes into account the values associated with the cultural context” and provide descriptions of why it is particularly relevant and important in working with Appalachian students. This paragraph can be improved by providing information on how the authors define a culturally responsive approach in this study. The existing definition seems too vague.

Although mindsets (Line 96) are important factors in retention and completion, this can potentially lead to passing the responsibility of low retention and completion rates to individual students – because they “believe” that STEM abilities are fixed and unchanging, they are low-performing. The authors might consider adding literature on trait-factor theories that aimed to match assumed stable personality traits to occupational characteristics. The issue is related to how the assessments were designed to match their personalities/abilities to occupations, not their belief system.

As the authors pointed out in Lines 147-153, it is unclear who should be providing mentorship. However, the authors could still benefit from providing how they define “mentors” and “mentorship” in this study at the end of the introduction (P. 4).

Two of the mentoring qualities – empathy and listening – are critical elements of SEL skills. SEL is a widely researched area, and there are many professional development resources available in relation to SEL. In the implication section, consider including ways to utilize these existing resources and adapt to the particular context of working with Appalachia students.

Author Response

(The authors gave the same response as above.)

Round 2

Reviewer 2 Report

Comments and Suggestions for Authors

I thank the authors for taking due consideration my comments. I have no further questions or concerns. 

Author Response

Thank you!